# Contribution of Outpatient Ultrasound Transvaginal Biopsy and Puncture in the Diagnosis and Treatment of Pelvic Lesions: A Bicenter Study

**DOI:** 10.3390/diagnostics13030380

**Published:** 2023-01-19

**Authors:** Irene Pelayo-Delgado, Javier Sancho, Mar Pelayo, Virginia Corraliza, Belen Perez-Mies, Cristina Del Valle, Leopoldo Abarca, Maria Jesus Pablos, Carmen Martin-Gromaz, Juan Ramón Pérez-Vidal, Inmaculada Penades, Elvira Garcia, Maria Carmen Llanos, Juan Luis Alcazar

**Affiliations:** 1Department of Obstetrics and Gynecology, University Hospital Ramón y Cajal, Alcalá de Henares University, 28034 Madrid, Spain; 2Department of Radiology, Hospital HM Puerta del Sur. Hospital HM Rivas, 28938 Madrid, Spain; 3Department of Pathology, University Hospital Ramón y Cajal, 28034 Madrid, Spain; 4Department of Obstetrics and Gynecology, University Hospital Virgen de la Arrixaca, 30120 Murcia, Spain; 5Department of Obstetrics and Gynecology, Clinica Universidad de Navarra, 31008 Pamplona, Spain

**Keywords:** tru-cut biopsy, core needle biopsy, transvaginal ultrasound, ultrasound-guided invasive procedures, fine-needle aspiration cytology, diagnosis

## Abstract

Background: The use of transvaginal ultrasound guided biopsy and puncture of pelvic lesions is a minimally invasive technique that allows for accurate diagnosis. It has many advantages compared to other more invasive (lower complication rate) or non-invasive techniques (accurate diagnosis). Furthermore, it offers greater availability, it does not radiate, enables the study of pelvic masses accessible vaginally with ultrasound control in real time, and it is possible to use the colour Doppler avoiding puncturing large vessels among others. The main aim of the work is to describe a standardized ambulatory technique and to determine its usefulness. Methods: This is a retrospective study of ultrasound transvaginal punctures (core needle biopsies and cytologies) and drainages of pelvic lesions performed on an outpatient basis during the last two years. The punctures were made with local anesthesia, under transvaginal ultrasound guidance with an automatic or semi-automatic 18G biopsy needle with a length of 20–25 cm and a penetration depth of 12 or 22 mm. The material obtained was sent for anatomopathological, cytological and/or microbiological study if necessary. Results: A total of 42 women were recruited in two centers. Fifty procedures (nine punctures, seven drains, and 34 biopsies) were performed. In five cases the punction and drain provided clinical relief in benign pelvic masses. Regarding material of the biopsies performed, 15 were vaginal in women previously histerectomized, finding 10 carcinomas, eight were ovarian tumours in advanced stages or peritoneal carcinomatosis obtaining the appropriate histology in each case, seven were suspicious cervical biopsies finding carcinomas in five of them, three were myometrial biopsies including one breast carcinoma metastasis in the miometrium and a benign placental nodule, and a periurethral biopsy was performed on a woman with a history of endometrial cancer confirming recurrence. The pathological diagnosis was satisfactory in all cases, confirming the nature of the lesion (25 malignant—ten vaginal recurrences of previous gynaecological cancers, eight cases of primary ovarian/peritoneal carcinoma, four new diagnosis of cervical malignant masses, one cervical metastasis of lymphoma, one periurethral recurrence of endometrial carcinoma and one recurrence of breast cancer in the myometrium—and 23 benign). The tolerance was excellent and no complications were detected. Conclusion: The ambulatory ultrasound transvaginal puncture and drainage technique is useful for obtaining a sample for pathological and microbiological diagnosis with excellent tolerance that can be used to rule out the recurrence of malignant lesions or progression of the disease, diagnose masses not accessible to gynecological exploration (vaginal vault, myometrium or cervix) and for early histologic diagnosis in cases of advanced peritoneal carcinomatosis or ovarian carcinoma as well as drainage and cytological study of cystic pelvic masses.

## 1. Introduction

Tru-cut biopsy, or core needle biopsy, and fine-needle aspiration cytology (FNAC) are minimally invasive ultrasound-guided techniques available to obtain tissue samples or cytology [1]. While Fine needle aspiration allows a cytological study of predominantly cystic lesions without solid tissue for biopsy [2], tru-cut biopsy allows not only histological study but also the immunohistochemistry of the tumour [3].

The use of transvaginal ultrasound as a guiding technique compared to CT or MRI has many advantages such as greater availability and easy choice of the moment of performance as it can be done at the same moment of the ultrasound, and besides, it does not radiate. The ultrasound is a dynamic process in which the examiner can see directly the movement or displacement of organs such as the intestinal handles, thus avoiding injury and it allows for the exact control when inserting the needle inside the pelvis. Transvaginal access enables the study of pelvic masses that by the conventional abdominal route would have a high risk of intestinal or vascular lesions, as well as nodules difficult to access by other routes. Performing the biopsy with ultrasound control in real time allows visualizing continuously the tip of the needle limiting the possibility of diagnostic errors or subsequent complications. In addition, the use of colour Doppler allows selecting the appropriate area to take the biopsy, avoiding puncturing large vessels or areas of intense vascularization. Other benefits include the short duration of the process and that it is more cost-effective [4].

These techniques are of great help in patients who present a lesion that requires definitive histological or cytological diagnosis for its management or for direct drainage of certain non-solid masses. While aspiration puncture provides information about cellular characteristics, biopsy allows tissue assessment by providing a tissue cylinder for histological study. Likewise, obtaining culture material allows microbiological study if necessary. In cases where malignancy is suspected in a pelvic mass, it is important to perform a histological study, since the management must be adapted to each specific tumour. In cases where a laparotomy or exploratory laparoscopy is required to obtain the biopsy, we have to take into account the need of hospital admission, general anaesthesia and increase not only costs but also morbidity and mortality from surgery as well as a delay in further treatment.

These techniques can be performed in outpatient clinic after the ultrasound evaluation of the patients, as they do not require any special preparation or general anaesthesia. They are therefore simple, safe and cost effective tools for the management of patients [5].

The main aim of the work is to describe concisely ambulatory echo-guided transvaginal puncture and drainage technique and to determine its usefulness in a series of patients, in order to establish its possible indications including not only pelvic masses but also suspicious vaginal, cervical, myometrial, adnexal, peritoneal or other pelvic lesions accessible to the transvaginal ultrasound approach.

## 2. Materials and Methods

This is a retrospective analysis of data prospectively collected of ultrasound transvaginal guided punctures (tru-cut biopsies and cytologies) and drainages of pelvic lesions performed on an outpatient basis during the last two years in two Departments of Obstetrics and Gynaecology of University Hospitals in Madrid and Murcia (Spain). We obtained approval from the Ethics Committee of each participant University Hospital.

Patients were selected by consensus in a multidisciplinary team discussion meeting, including gynaecological and medical oncology departments, after an initial ultrasound evaluation. Baseline status, clinical history and chronic treatments were evaluated in order to assess the safety of the test. The patient was informed and signed the informed consent document for ultrasound-guided transvaginal puncture. Patients were excluded if the mass was not accessible by transvaginal ultrasound without injuring large vessels, bladder or bowel loops. All cases in which ultrasound biopsy of both vaginal and cervical nodules was performed were not accessible or not seen with speculoscopy

Table 1 shows the indications for the procedure.

The material used included a biopsy guide attached to the transvaginal probe (5–7.5 MHz intracavitary transducer, GE Voluson E8), a puncture/drainage needle and an automatic biopsy device. The puncture/drainage needle is an 18G, 25 cm long (Bard Magnum ^®^) attached to an adapter to insert the suction syringe comfortably (Figure 1).

The biopsy is performed under ultrasound control using an automatic or semi-automatic 18G biopsy needle with a length of 20–25 cm and a penetration depth of 12 or 22 mm (Biopince ™Utra/ABG-2020) (Figure 2).

The amount of cylinders collected varies from 1–3, to ensure a sufficient amount of tissue that allows the pathologist to correctly study the sample. Both invasive techniques and ultrasound scans were carried out by ultrasound experts.

The technique was performed on an outpatient basis in all cases. Before the procedure, 2 g oral azithromycin was administered. The patient is placed in lithotomy position and with the help of a speculum the vagina is aseptized with aqueous chlorhexidine or povidone iodine. The area is then sprayed with lydocaine spray (10 mg/pulsation). Additionally, in some cases, 10 mL of local anaesthesia (mepivacaine 2%) is instilled into the four quadrants of vaginal sac bottom with a spinal needle 22G 90 mm long. The optimal site for puncture/drainage is located by direct visualization of the lesions and with the help of the Doppler those areas with increased vascularization should be avoided. Using the reference guide attached to the transvaginal probe, the procedure is performed (drainage/puncture/biopsy) (Figure 3). The material obtained is sent for pathological, cytological and/or microbiological study if necessary.

To ensure the suitability of the obtained material for diagnosis, rapid on site evaluation (ROSE) [6] is performed by a pathologist present in the examination room. The obtained core tissue is transferred from the needle to a clean slide and gently pushed and rubbed down with another slide to obtain a cytology specimen. After that, the core is immediately fixed in 10% buffered formalin and the cytology specimen is dyed with Diff-Quick to evaluate its suitability for diagnosis and to make an initial diagnostic approach (Figure 4). In case of adequate ROSE, additional passes (up to a total of three) are collected and placed directly in formalin to assure the possibility of additional immunohistochemistry or molecular studies. Otherwise, the procedure will be repeated up to a maximum of three passes. The pathological diagnosis was satisfactory if the pathologist obtained enough material for histological analysis and immunohistochemistry if necessary, or the aspirated liquid was not contaminated and was suitable for the cytological exam and/or culture. The agreement with the definitive pathological anatomy was verified in the cases in which the process was completed with subsequent surgical treatment.

A correct haemostasis is ensured, sometimes requiring the application of a small amount of solution of ferrous sulphate in the vagina. After a period of observation of 15–30 min, the patient is discharged with the recommendations to go to the Emergency Department of the Reference Hospital in the case of fever, intense abdominal pain, malodorous discharge or severe bleeding. After 1–2 weeks, an appointment is scheduled to report the result of the sample obtained and discuss further treatment. Possible complications were reported: Vaginal bleeding, blood transfusion, hospital admission, urgent surgery, pelvic infection, sepsis or death.

Table 2 shows a description of the steps of the technique.

We recorded the patient characteristics, including age, menopausal state (pre/postmenopausal), personal history of cancer (ovarian, endometrial, uterine, cervical, vaginal, breast cancer or other non-gynaecological malignancy), as well as ultrasound features of the punctured masses such as lesion type (solid, solid-cystic, cystic), maximum diameter of the lesion, evaluation of Doppler colour (score color 1–4), localization (ovary, uterus, cervix, vaginal, peritoneum, and other locations).

## 3. Results

A total of 42 women were included and 50 procedures (nine punctures, seven drains, and 34 biopsies) were performed (Table 3). Eight women had two procedures (repeated drain or puncture)

The mean age of the patients was 56.2 years (standard deviation: 13.3; ranging from 25 to 79 years), 33 of them were menopausal (80.49%). Mean Body mass index was 28.7 (standard deviation: 6.7; ranging from 19.4 to 41.0). Twenty-six percent of patients had some type of obesity.

Twenty-five patients had a personal history of cancer, all of them gynaecological (nine endometrial adenocarcinoma, five ovarian carcinoma, five cervical cancer, two breast cancer, one uterine sarcoma and one tubal cancer) except for one case of previous melanoma and a non-Hodgkin Lymphoma. Twenty-three patients (56.1%) had a previous abdominal surgery (eighteen hysterectomy and bilateral salpingo-oophorectomy, three anexectomy, one hysterectomy and one myomectomy). A total of 23 patients had treatment with pelvic radiation therapy (n: 8, 19.5%) and/or chemotherapy (n: 15, 36.6%).

The indication for the procedure was to provide histopathological diagnosis in 40 cases (additionally, clinical relief was seek in three cases were large cystic masses in women with significant co-morbidites that precluded surgical intervention) and drainage of tubo-ovarian abscess in two cases.

Regarding the type of lesion found in transvaginal ultrasound, 32 were solid masses (80.0%), and eight were cystic-solid masses (20.0%), with a size ranging between 10 and 128 mm (mean size: 47.5 mm; standard deviation: 30.9). Moderate-intense vascularization (Doppler score colour three–four) was present in 24 of these masses (60.0%). We performed a vaginal nodule puncture in 15 cases (38.5%, 15/39), eight non-operable ovarian masses (20.5%), seven were cervix uterine masses (17.5%), five uterine corpus suspicious lesions (12.5%), three peritoneal masses (7.5%), one tubal drainage and one periurethral nodule.

The histopathological study was performed in all cases, except in two cases of punction and drainage of tubo-ovarian inflammatory pelvic disease. Among the samples studied (N: 40), we found malignancy in 25 cases (62.5%). We found ten cases of vaginal recurrences of previous gynaecological cancers (six endometrial adenocarcinoma, three cervical carcinoma and one leiomiosarcoma) as well as one cervical metastasis of lymphoma, one periurethral recurrence of endometrial carcinoma and one recurrence of breast cancer in the myometrium. Additionally, seven cases of primary ovarian carcinoma (five high-grade serous ovarian carcinoma, one carcinosarcoma and one endometrioid adenocarcinoma) and four new diagnoses of cervical masses (one non-Hodgking lymphoma, one atypical polypoid adenomyoma, one urothelial carcinoma and one sarcoma) and one case of peritoneal cancer were also diagnosed. Finally, we diagnosed one nodule of a benign placental site within the myometrium.

As result of the biopsy study, 31 complementary treatments were performed, such as antibiotic therapy (n: three cases), radiotherapy (n: five cases), chemotherapy (n: 13 cases), and complementary surgery (n: ten cases). The histological study of the patients undergoing surgery was the same as that obtained in the biopsy (accuracy 100%), including one case of endometrial glandular atypia in the biopsy whose histological definitive result was carcinoma.

Regarding the subsequent follow-up of the patients, only one patient went to the emergency room five days after the process with febricula not attributed to the technique. Another patient died seven days after the procedure due to the evolution of a previous encephalitis.

The tolerance of the technique was excellent and all of the patients were discharged 15–20 min after the procedure. No complications were detected in any of the cases during the procedure.

## 4. Discussion

Fine needle aspiration cytology can be used for cytological examination of nonsolid lesions to rule out malignant pathology or recurrence diagnosis in patients with personal history of gynecological tumours with a sensitivity of 88% and specificity of 88%, as well as drainage of symptomatic nonsolid lesions [7]. Since the 1990s, FNAC of benign pelvis masses has shown to be an easy method to perform and is simple and safe [2,8,9,10]. However, it should not be considered in all cases of benign adnexal masses given its high rate of recurrence, which can be up to 63.5% [2].

In our study, we performed puncture and drainage in nine patients. In two cases, the presence of ascites was cytologically studied in a woman with a history of ovarian cancer and in another woman with a previous mucinous cystadenocarcinoma with a cystic lesion in the pouch of Douglas. Other three cases were cystic symptomatic lesions and 2 diagnosed of pelvic inflammatory disease, all of them with a significant clinical improvement after drainage, as it is described in bibliography [11], even though one of the cystic lesions recurred in the posterior control but persisted asymptomatic. In all these cases, the cytological study was benign. In only one patient there was a positive cytology also confirmed by biopsy in a peritoneal carcinomatosis. Therefore, we can conclude that this technique could be used in selected cases with sometimes-symptomatic nonsolid pelvic masses in patients with high surgical risk, to avoid surgery for direct cytological diagnosis of benign lesions and relieving the patient’s clinic.

Taking a core needle biopsy allows more tissue to be obtained than in a cytology [12], with a preserved tissue sample architecture, and immunohystochemical studies can be performed increasing the diagnostic rate. In our study, all pathological diagnoses were satisfactory being sufficient for treatment planning, in accordance with the definitive histological result in the patients in whom the surgical intervention was subsequently performed. The nine cases in which the patients had complementary surgery, the histological definitive study of the patients was the same as that obtained in the biopsy (accuracy 100%). There was a 78-year-old patient with a previous diagnosis of malignant melanoma in the popliteal hollow, with a hysterectomy and bilateral salpingo-oophorectomy for previous benign pathology, who presented a lesion in the vaginal vault not visible to the gynaecological examination with speculum. The biopsy diagnosis was of glandular atypia and the histological definitive result was carcinoma. Other authors have described failed diagnosis from 1% [13] to 1.7% [4,14,15], which could be related to cystic or necrotic tumours.

Regarding the accuracy of the technique, the study carried out by Fisherova et al. [3] was one of the first to demonstrate the diagnostic accuracy of ultrasound-guided core needle biopsies obtained vaginally and abdominally (n: 86) that reached 97.7%. Two years later, Zikan et al. [4] also took a group of 195 abdominal and vaginal biopsies of which had definitive histological confirmation in 118 patients who underwent subsequent surgery with an accuracy of 98.3%. Verschuere et al. [5] analyzed 176 tru-cut biopsies obtained vaginally. In their case, when taking at least two cylinders, diagnostic adequacy increased to >95% and compared to final histology, the diagnostic accuracy was 97.2%. Buonomo et al. [15] recently analyzed 44 abdominal and vaginal biopsies in which they obtained a diagnostic accuracy of 88.2%. However, considering only the cases with at least two diagnostic samples, accuracy rose to 94.1%.

We obtained an average of two samples (one–three) with an accuracy of 100% similar to what is described in the literature [3]. To increase biopsy performance, it is advisable that a pathologist reviewed the sample obtained at the time of extraction (ROSE method) so that, in case of obtaining a scarce sample, the biopsy can be repeated to obtain more material. The immediate assessment by the pathologist decreases the risk of a false-negative result in the case of scanty material [16]. In addition, in cases of very aggressive cancers, you must be sure that you have not taken only tumor necrotic tissue and the sample is valid for histological analysis.

One of the great advantages of the technique is that it can be performed in an outpatient basis, with no need of special operating rooms or requiring subsequent admission. In our experience, after the procedure, all the women were discharged, and continued with their usual activity, avoiding work absence. With regard to anaesthesia, most studies suggest that it is not necessary. We used some local anaesthesia to decrease the puncture sensation and increase the patient’s comfort. Plett et al., in their review, suggested the instillation of 1 to 3 mL of 1% lidocaine prior to the puncture using a 22-gauge needle inserted via the needle guide and under direct sonographic visualization. We prefer to put it under direct visualization in the vagina with the speculum to increase the chances of anesthetize the entire vaginal area and not only a specific point, which would benefit in the case of having to perform several punctures [17].

The rates of complications recorded in the literature are scarce and of little importance [1]. However, special attention should be paid to the general baseline condition of the patients, and rule out thrombocytopenia, which some authors consider to be a contraindication for the process [3,4] because of an increased risk of further bleeding. On the other hand, to decrease the risk of subsequent infectious complications, it is highly recommended to clean the vagina prior to biopsy [18]. We used chlorhexidine or iodized povidone and had no complications related to infection from the biopsy site. Neither the body mass index, which reached up to 41.0 (type II obesity, 26.2% of patients had some type of obesity), nor previous pelvic surgeries (in a 56.1% in our series), nor pelvic radiation therapy (19.5%) and/or chemotherapy (36.6%) influenced our results.

Another important tip to reduce complications is to calculate the exact point of the biopsy. In cases of biopsy by automatic tru-cut biopsy device, it is advisable to measure the distance from the transvaginal probe inside the pelvis to the lesion under study compared to the tip of the biopsy disposal and the shooting distance, in order to lessen injuries to adjacent tissues (vessels, intestine, and bladder). In addition, the application of colour Doppler can help identify vascular areas to try to avoid. If after the puncture there is a small intralesional bleeding detected by Doppler, it is advisable to maintain the pressure with the transvaginal probe to stop it. In our series, only one patient went to the emergency room five days after the process. She was a 31-year-old woman with an extensive peritoneal carcinomatosis with liver metastases who attended the emergency department for febricula (37.5 °C). No signs of abdominal infection were detected either in the laboratory study or in the complementary imaging tests, so it was classified as fever of unknown origin related to carcinoma. Another patient died seven days after the procedure. She was a 59 years old woman previously hospitalized in the neurology department with a severe encephalopathy of sudden onset. Imaging tests revealed an incidental ovarian mass of 85 mm highly suspected of malignancy whose pathological diagnosis was carcinosarcoma.

In relation to the vaginal drainage of tube-ovarian abscesses, it is an established technique and some authors have stated that transvaginal ultrasound-guided aspiration combined with antibiotics is a first-line procedure because it is an effective and safe management [19]. Chong et al. obtained a clinical success rate for tuboovarian abscesses, pelvic collections and overall pelvic collections were 83%, 90% and 88%, respectively [20]. Recently, ultrasound-guided trans-uterine cavity core needle biopsy has been proposed as a method for discriminating uterine benign leiomyomas from sarcomas with very interesting results [21].

It is important to emphasize the great benefit presented by the tru-cut technique in the histological diagnosis of patients with peritoneal carcinomatosis or advanced adnexal carcinomas [22], as a biopsy is required to determine the nature of the tumour and perform a treatment appropriate to each histological type. In these cases it is especially indicated to take this type of biopsy provided that we have material accessible with the vaginal probe, that prevents the performance of surgery (laparoscopic or laparotomic) with all associated morbidity and mortality, with no need of preparatives prior to surgery (chest x-ray, electrocardiogram, visit to pre-surgical anaesthesiologist), no hospitalization and recovery days, and allows the initiation of chemotherapy and/or radiation therapy faster in a the patient in better physical conditions. It has also influence in the delay in the diagnosis taking into account the days prior to surgery and the recovery of the patient. We found seven cases of ovarian carcinoma (five high-grade serous ovarian carcinoma, one carcinosarcoma and one endometrioid adenocarcinoma) and one case of peritoneal carcinomatosis. In all cases, chemotherapy appropriate to each type of tumour could be started early.

This study reflected the ambulatory technique of puncture and biopsy of pelvic masses that can be used vaginally, establishing the indications for it. However, its generalization may present limitations given the reduced sample, which has only taken place in two different centres, and that the ultrasound and the technique has been done by experienced ultrasound operators. Other research studies could include more centres and more sonographers with different skills and experience [11].

## 5. Conclusions

The ambulatory echo-guided transvaginal puncture and drainage technique described is useful both for obtaining a sample for pathological diagnosis (cytology and biopsy) and microbiological (culture material), as for the treatment of certain pelvic lesions (drainage) with excellent tolerance by patients. It has been shown to be beneficial both in women with a history of previous oncological processes to rule out recurrence or progression of the disease, and for the diagnosis of lesions not visible on examination (peritoneum, vaginal vault, myometrium or cervix). It also allows for early histological diagnosis in cases of peritoneal carcinomatosis or advanced ovarian carcinoma, facilitating the introduction of personalized neoadjuvant chemotherapy for the patient without requiring initial surgery.

## Figures and Tables

**Figure 1 diagnostics-13-00380-f001:**
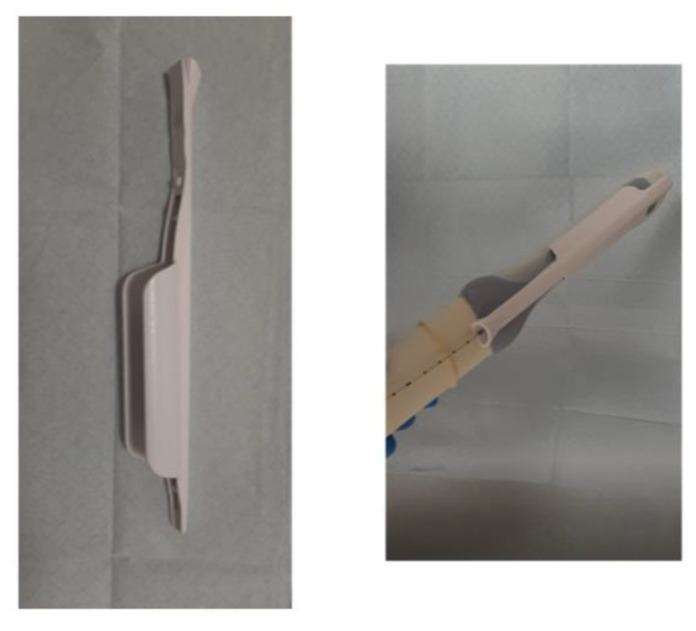
Biopsy guide attached to the transvaginal probe through which a needle or puncture device is inserted.

**Figure 2 diagnostics-13-00380-f002:**
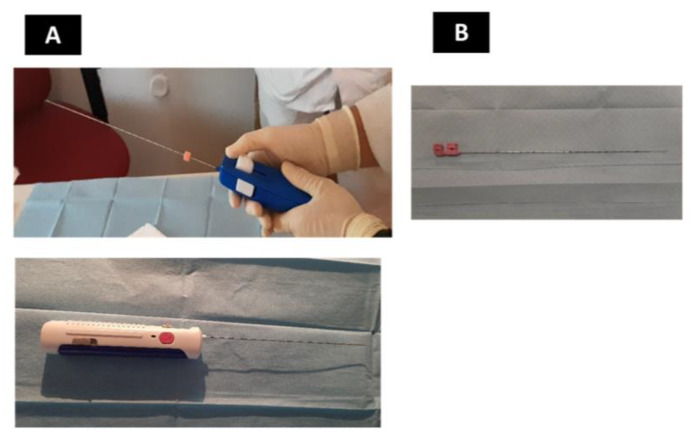
Puncture/drainage needle (**A**) and 18G automatic or semi-automatic biopsy device with a length of 20–25 cm and a penetration depth of 12 or 22 mm (**B**).

**Figure 3 diagnostics-13-00380-f003:**
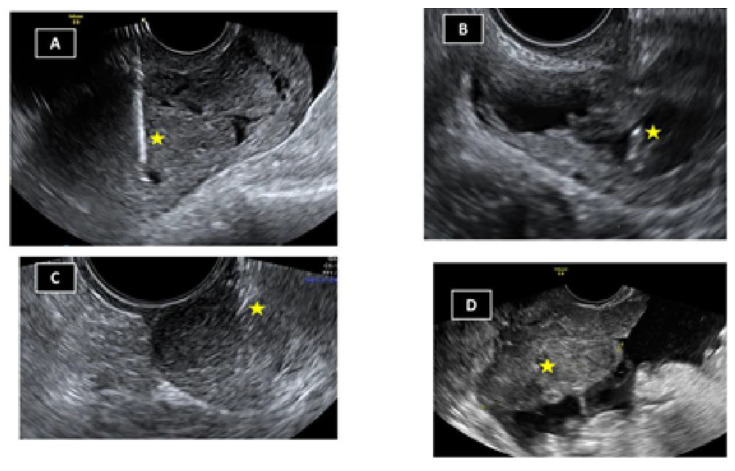
Examples of transvaginal ultrasound punctures/drainages/biopsies: (**A**) Myometrium; (**B**) tubal abscess; (**C**) vaginal nodule; (**D**) ovarian tumor with peritoneal carcinomatosis. Yellow asterisk: Needle.

**Figure 4 diagnostics-13-00380-f004:**
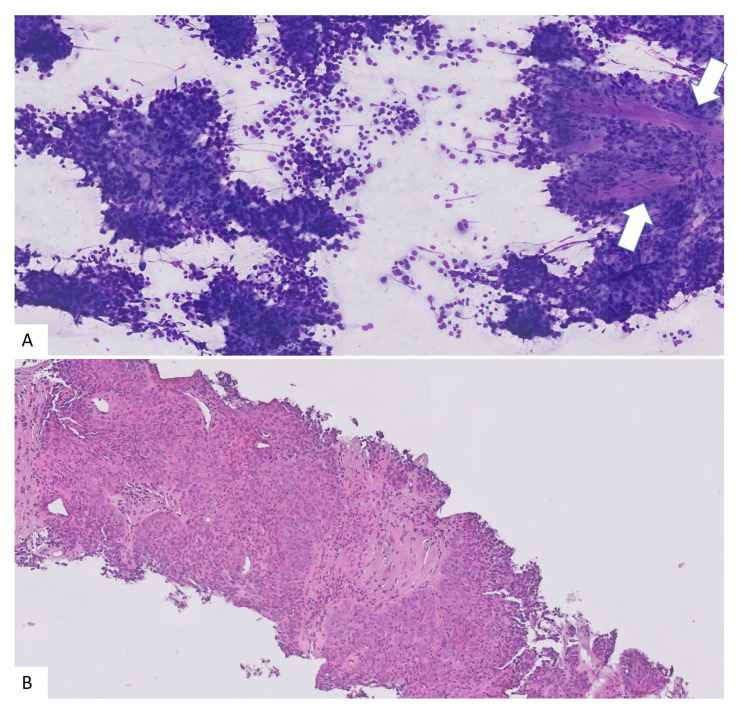
(**A**) Cytology obtained for ROSE. High cellularity present in the slide, with papillary architecture (Arrows). The specimen was considered adequate and a high-grade carcinoma was suspected (Diff-Quick 10×). (**B**) The correspondent core biopsy after histological processing showing a high-grade serous carcinoma (H&E 10×).

**Table 1 diagnostics-13-00380-t001:** Indications for transvaginal echo-guided puncture.

The ambulatory ultrasound transvaginal puncture and drainage technique is useful for: - obtaining a sample for pathological (cytology and biopsy) and microbiological (culture material) diagnosis - treatment of pelvic lesions (drainage) It can be used in cases of: - women with a history of previous oncological processes to rule out recurrence or progression of the disease - diagnosis of lesions not accessible to gynecological exploration (vaginal vault, myometrium or cervix) - early histologic diagnosis in cases of advanced peritoneal carcinomatosis or ovarian carcinoma, facilitating the introduction of personalized neoadjuvant chemotherapy for the patient without requiring initial surgery

**Table 2 diagnostics-13-00380-t002:** Summary of steps to be taken for performing transvaginal echo-guided puncture.

Selection of the patient: evaluate the accessibility of the pelvic tumor transvaginally and the clinical conditions Prepare the transvaginal probe with the ultrasound gel, the cover, and the biopsy guide, making sure it is well aligned with the transvaginal tube Place the speculum, clean with aqueous chlorhexidine or povidone iodine Instill local anesthesia into the 4 quadrants of the vaginal sac bottom by rotating the speculum Remove the speculum and insert the transvaginal probe with the needle or biopsy hidden inside the biopsy guide attached to the transvaginal probe Ultrasound location of the optimal puncture/drainage site using the transvaginal probe-coupled reference guide Perform the puncture, biopsy or drainage. Make sure you take at least two biopsy cylinders if possible. Place the speculum again and check for hemostasis (use of ferrous sulphate if necessary) After observation of 15–30 min with the patient seated, she can go home Recommendations: If fever, intense abdominal pain, malodorous discharge or severe bleeding you should go to the Emergency Department of the Reference Hospital

**Table 3 diagnostics-13-00380-t003:** Description of the procedures performed.

Procedure	N	Indication	Biopsy/Culture
Puncture and drainage	9	Symptomatic pelvic cyst lesion in vaginal vault (peritoneum/inflammatory pelvic disease)	Benign/1 Gram + 1 Adenocarcinoma
Myometrial biopsy	3	Rule out metastasis/suspicious lesions	Benign (myoma) Metastasis breast carcinoma Benign placental nodule
Vaginal vault biopsy	15	Previous hysterectomy (carcinoma)	5 Recurrent endometrial cancer 3 Recurrent cervical cancer 1 Recurrent leiomyosarcoma 1 Glandular atypia/carcinoma 5 Benign
Cervical biopsy	7	Suspicious cervical mass	1 Non-Hodgkin lymphoma 1 Recurrent non-Hodgkin linfoma 1 Atypical polypoid adenomyoma 1 Sarcoma 1 Urothelial carcinoma 1 Benign 1 Recurrent adenocarcinoma
Ovarian/peritoneal mass biopsy	8	Suspected nonoperable ovarian tumor/peritoneal carcinomatosis	5 Ovarian high grade serous carcinoma 1 Adenocarcinoma 1 Carcinosarcoma 1 Peritoneal cancer
Other	1	Periuretral nodule	Recurrent endometrial cancer

## Data Availability

Data are available upon reasonable request.

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
