# Peer review of "Contribution of Outpatient Ultrasound Transvaginal Biopsy and Puncture in the Diagnosis and Treatment of Pelvic Lesions: A Bicenter Study"

_diagnostics, 2023, doi:10.3390/diagnostics13030380_

Round 1

Reviewer 1 Report

Title: rather than multicenter, put bicenter

Abstract

Background a little too succinct

42 patients for how many centres?

Synthesize the anatomopathological results.

Introduction

P2 L71-75: what is the place of biopsies in case of suspected sarcoma?

Materials and methods

P4 A136: correct the sentence "it's important to locate the optimal site"

P4 L146: what is the benefit of the pathologist being present in the examination room? doesn't it complicate the procedure? this is stated on P8 L292, but is it really necessary?

Results :

Age, BMI: put only 1 decimal place

Likewise the other percentages in the paragraph, put only one decimal place

Vaginal nodule puncture in 15 cases (P7 L229): was it not accessible to the speculum? As were the 7 cervical masses (P7 L230)?

Figure 1 and 2 are they really useful?

References

These are well written and referenced.

It seems to me that the discussion could be more substantiated and that some articles are missing and could feed the discussion which I find a bit simple and lacking in comparison with already existing studies/articles:

- Regarding the procedure: Plett SK et al. 2016

- Concerning tuboovarian abscesses: Gjelland K et al. 2005, Lebreton A et al. 2020

- Regarding pelvic collections: Chong LY et al. 2016

- Concerning lesions: Eitan R et al. 2017

In total:

This is an interesting study evaluating on 42 patients (50 procedures), the feasibility of diagnosis by transvaginal ultrasound-guided puncture.

A summary table of the indications and procedures according to the patients would have been useful.

Indeed, it is not clear why some patients had several procedures (2? 3?).

In addition, it is perhaps necessary to distinguish between diagnoses for bacteriological or anatomopathological purposes.

In the title, the term "treatment" is also mentioned. This only concerns 2 or 3 patients... so should we really talk about it?

Also, please clarify why some vaginal or cervical nodules were not biopsied under speculum.   

Author Response

Reviewer 1

  1. Title: rather than multicenter, put bicenter

Response: Done as suggested

  1. Abstract
  2. Background a little too succinct

Response: Sentences added

  1. 42 patients for how many centres?

Response: data added

  1. Synthesize the anatomopathological results.

Response: modified as suggested

  1. Introduction
  2. P2 L71-75: what is the place of biopsies in case of suspected sarcoma?

Response: Thanks for this comment. We added a sentence and reference in the Discussion

  1. Materials and methods
  2. P4 A136: correct the sentence "it's important to locate the optimal site"

Response: corrected. Sentence re-worded

  1. P4 L146: what is the benefit of the pathologist being present in the examination room? doesn't it complicate the procedure? this is stated on P8 L292, but is it really necessary?

Response: We do believe the presence of a pathologist can decrease of false negative results. Sentence added as well as a reference

  1. Results:
  2. Age, BMI: put only 1 decimal place

Response: Done as suggested

  1. Likewise the other percentages in the paragraph, put only one decimal place

Response: Done as suggested

  1. Vaginal nodule puncture in 15 cases (P7 L229): was it not accessible to the speculum? As were the 7 cervical masses (P7 L230)?

Response: Actually not, in all these cases the lesion were submucosal, palpable, but not visible by speculum. Sentence added

  1. Figure 1 and 2 are they really useful?

Response: we do think they could be interesting for the reader. We suggest not deleting them

  1. References: These are well written and referenced.

Response: thanks

  1. It seems to me that the discussion could be more substantiated and that some articles are missing and could feed the discussion which I find a bit simple and lacking in comparison with already existing studies/articles:Regarding the procedure: Plett SK et al. 2016. Concerning tuboovarian abscesses: Gjelland K et al. 2005, Lebreton A et al. 2020. Regarding pelvic collections: Chong LY et al. 2016.Concerning lesions: Eitan R et al. 2017

Response. Thanks for this comments. Additional text added with reference to these studies

  1. In total: This is an interesting study evaluating on 42 patients (50 procedures), the feasibility of diagnosis by transvaginal ultrasound-guided puncture.

Response. Thanks for this comment

  1. A summary table of the indications and procedures according to the patients would have been useful.

Response: table added

  1. Indeed, it is not clear why some patients had several procedures (2? 3?).

Response: some patients had more than one procedure. Sentence added.

  1. In addition, it is perhaps necessary to distinguish between diagnoses for bacteriological or anatomopathological purposes.

Response: data clarified. See new table

  1. In the title, the term "treatment" is also mentioned. This only concerns 2 or 3 patients... so should we really talk about it?

Response: we believe that the term treatment refers not only to patients to whom the drainage of the lesion serves as a direct treatment but that the fact of having a histological diagnosis contributes to the establishment of an early and accurate treatment to patients to whom the diagnose of a cancerous lesion or lesion susceptible to further treatment is available

  1. Also, please clarify why some vaginal or cervical nodules were not biopsied under speculum.   

Response. Information added

Reviewer 2 Report

The present study describes a standardized ambulatory technique to perform transvaginal ultrasound guided biopsy and puncture of pelvic lesions. The procedure is clearly described, making it reproducible. The second aim is to determine the usefulness of the procedures, even if there are many articles in the literature with larger population that already showed it.

Study design is not described clearly. The article is presented as a prospective work in the abstract (line 24) and as a retrospective analysis of data prospectively collected in the text (line 90). Reading the methods, it is described more like as an analysis of routinely collected data.

The work is interesting as it propones the use of ROSE, despite the demonstrated diagnostic efficacy of the procedure.   

A control group would be important to understand its usefulness, and data on its cost-effectiveness should be presented.

Did you also perform a MOSE (Macroscopic On-Site Evaluation) for histological samples?

It would be very interesting to evaluate the impact of MOSE on diagnostic accuracy of transvaginal ultrasound guided biopsy.

As reported by the authors, many studies showed that anesthesia is not necessary, but they suggest instilling local anesthesia. Also in this case, a control group could be useful to prove cost-effectiveness of the standardized ambulatory technique proposed. 

Author Response

Reviewer 2

  1. The present study describes a standardized ambulatory technique to perform transvaginal ultrasound guided biopsy and puncture of pelvic lesions. The procedure is clearly described, making it reproducible. The second aim is to determine the usefulness of the procedures, even if there are many articles in the literature with larger population that already showed it.

Response: Thanks for this comment. No change made

  1. Study design is not described clearly. The article is presented as a prospective work in the abstract (line 24) and as a retrospective analysis of data prospectively collected in the text (line 90). Reading the methods, it is described more like as an analysis of routinely collected data.

Response: Sorry for this. This is actually a retrospective analysis of prospectively collected data in our institutions about a certainly routine procedure. Abstract modified

  1. The work is interesting as it propones the use of ROSE, despite the demonstrated diagnostic efficacy of the procedure.   

Response: Thanks for this comment. Yes, we stress this approach. Reference added for support our concept

  1. A control group would be important to understand its usefulness, and data on its cost-effectiveness should be presented.

Response: This is a very interesting comment. However, unfortunately, we do not have a control group and we cannot present data about cost-effectiveness

  1. Did you also perform a MOSE (Macroscopic On-Site Evaluation) for histological samples? It would be very interesting to evaluate the impact of MOSE on diagnostic accuracy of transvaginal ultrasound guided biopsy.

Response: MOSE is proposed as an alternative if no pathologist is available.  We performed MOSE and ROSE but we gave more importance to ROSE because as you have a cytological examination, this allows to better guide the diagnosis and not only inform about the validity of the material (MOSE).

  1. As reported by the authors, many studies showed that anesthesia is not necessary, but they suggest instilling local anesthesia. Also, in this case, a control group could be useful to prove cost-effectiveness of the standardized ambulatory technique proposed. 

Response: Thanks for this comment. Plett et al in their review suggested the instillation of 1 to 3 mL of 1% lidocaine prior to the puncture using a 22-gauge needle inserted via the needle guide and under direct sonographic visualization. We prefer to put it under direct visualization in the vagina with the speculum to increase the chances of anesthesia of the entire vaginal area and not only a specific point, which would benefit in the case of having to perform several punctures. A sentence added.

Reviewer 3 Report

diagnostics-2138691-peer-review-v1

Manuscript title: Contribution of outpatient ultrasound transvaginal biopsy and puncture in the diagnosis and treatment of pelvic lesions. A multicenter study.

2023-01-01

A peer review.

The manuscript presents original research evaluating the usefulness of ultrasound-guided transvaginal punctures of pelvic lesions in a series of patients prospectively collected in two outpatient gynecology departments. It is a retrospective analysis of data about the clinical application of needle punctures of different lesions detected and targetable by ultrasonography. Among indications were solid lesions suspected for malignancy, cystic lesions presumed benign but symptomatic, and tubo-ovarian abscess, all accessible by transvaginal ultrasonography, thus this report represents a real-life setting. This issue is the main strength of this study. There are many descriptions and tips, which are clinically useful. The manuscript has high educational value because of detailed descriptions and presentations of the procedures – meticulous preparation to the procedure, anesthesia, the performance of the procedure, and checking the quality of obtained material (e.g. ROSE). Other strengths of the study are that there was no patients selection, and the data were prospectively collected and performed in two independent institutions. The text is well written, in the correct order, and with the text flow, making the reading easy and understandable.

There are however some issues that need corrections or clarification:

1.       I would suggest adding the term “core needle biopsy” to the list of keywords and maybe adding this term in the main text and in the abstract. It is because the “core needle biopsy” is used by the majority of physicians when the biopsy is performed under imaging control to obtain tissue for histological diagnosis. The “tru-cut biopsy” must be used because the majority of gynecological research on the subject uses this term.

2.       Line 51: not “Fine” but “fine”.

3.       Line 57: not “it’s” but “is”.

4.       Table 1 needs some rephrasing. If one presents a list of indications for transvaginal echo-guided biopsy, then words like “with excellent tolerance” should not be used. These words are a description of the results, and should not be placed in the list of indications.

5.       Line 114: there is some mistake. The sentence begins with “MHz” without and numerical value. Moreover, the sentence later does not make sense.

6.       Line 125-127: do the 3 figures really represent the description provided by the legend of Figure 2? To me, it looks like all 3 figures present a system for a tru-cut (core needle) biopsy performance. I cannot see the puncture /drainage needle (to which a suction syringe could be attached comfortably (as described in line 116)).

7.       Line 131: “Previously”? Should not rather “Before the procedure” be used?

8.       I suggest Authors consider changing the way of presenting the number of patients in each subgroup. The “N:” seems to me strange. In most of the studies, “n=” is rather used to present the number of cases in each subgroup. The “N:” looks like a part of some statistical analysis, and makes the reader confused.

9.       Line 223: not “women significant” but “women with significant”.

10.   Line 230”: not “uterine cervical masses”, but rather “uterine cervix masses”, or lesions not masses. And the following: “uterine … lesions” – if you separately list uterine cervix, then you should separately name uterine corpus because both, cervix and corpus represent parts of the uterus.

11.   Line 247: I would argue that the accuracy was 100% if you write that, the histology of the tru-cut biopsy was glandular atypia, and the histology from the definitive surgery was carcinoma. This is just the scientific and methodological issue of how you define accuracy. The most important is the clinical 100% usefulness of the core needle biopsy because clinically the management of both glandular atypia and carcinoma would be the same.

12.   I would encourage the Author to discuss more the accuracy of the histology obtained by the tru-cut biopsy as compared with the histology obtained from definitive surgery. It is a clinically very important issue. The reported 100% accuracy is great, however, one should elaborate on this issue more. The 100% accuracy may be attributable to the meticulous preparation and double-checking of the obtained material, as presented by the Authors in the manuscript (e.g. three samples (not 1-2), the ROSE protocol). It would be reasonable to compare results with such research like Gynecol Oncol 2011;161:845-851 (that Authors already cite) or J Clin Med 2022;11:2534 or those not so recent e.g. by Zikan and Fischerova.

13.   Line 259: The abbreviation of the FNAC was already explained earlier in the text (line 49), so you do need to repeat it here.

14.   Line 337: “found seen cases”? It seems like a language mistake.

Given the fact that some remarks are about English language issues, and me not being a native English speaker, I would recommend an editing of the text by a native English speaker.

In conclusion, I consider the manuscript very interesting, scientifically well conducted, and written, with additional high practical and educational value. I recommend acceptance of the manuscript after responding to the above-listed remarks. 

Author Response

Reviewer 3

  1. The manuscript presents original research evaluating the usefulness of ultrasound-guided transvaginal punctures of pelvic lesions in a series of patients prospectively collected in two outpatient gynecology departments. It is a retrospective analysis of data about the clinical application of needle punctures of different lesions detected and targetable by ultrasonography. Among indications were solid lesions suspected for malignancy, cystic lesions presumed benign but symptomatic, and tubo-ovarian abscess, all accessible by transvaginal ultrasonography, thus this report represents a real-life setting. This issue is the main strength of this study. There are many descriptions and tips, which are clinically useful. The manuscript has high educational value because of detailed descriptions and presentations of the procedures – meticulous preparation to the procedure, anesthesia, the performance of the procedure, and checking the quality of obtained material (e.g. ROSE). Other strengths of the study are that there was no patients selection, and the data were prospectively collected and performed in two independent institutions. The text is well written, in the correct order, and with the text flow, making the reading easy and understandable.

Response: Thanks for this comment.

  1. I would suggest adding the term “core needle biopsy” to the list of keywords and maybe adding this term in the main text and in the abstract. It is because the “core needle biopsy” is used by the majority of physicians when the biopsy is performed under imaging control to obtain tissue for histological diagnosis. The “tru-cut biopsy” must be used because the majority of gynecological research on the subject uses this term.

Response: Done as suggested

  1. Line 51: not “Fine” but “fine”.

Response: Corrected

  1. Line 57: not “it’s” but “is”.

Response: Corrected

  1. Table 1 needs some rephrasing. If one presents a list of indications for transvaginal echo-guided biopsy, then words like “with excellent tolerance” should not be used. These words are a description of the results, and should not be placed in the list of indications.

Response. Modified as suggested

  1. Line 114: there is some mistake. The sentence begins with “MHz” without and numerical value. Moreover, the sentence later does not make sense.

Response: Corrected

  1. Line 125-127: do the 3 figures really represent the description provided by the legend of Figure 2? To me, it looks like all 3 figures present a system for a tru-cut (core needle) biopsy performance. I cannot see the puncture /drainage needle (to which a suction syringe could be attached comfortably (as described in line 116)).

Response: Figure’s legend modified

  1. Line 131: “Previously”? Should not rather “Before the procedure” be used?

Response: Modified as suggested

  1. I suggest Authors consider changing the way of presenting the number of patients in each subgroup. The “N:” seems to me strange. In most of the studies, “n=” is rather used to present the number of cases in each subgroup. The “N:” looks like a part of some statistical analysis, and makes the reader confused.

Response: Modified as suggested

  1. Line 223: not “women significant” but “women with significant”.

Response: Modified as suggested

  1. Line 230”: not “uterine cervical masses”, but rather “uterine cervix masses”, or lesions not masses. And the following: “uterine … lesions” – if you separately list uterine cervix, then you should separately name uterine corpus because both, cervix and corpus represent parts of the uterus.

Response: Modified as suggested

  1. Line 247: I would argue that the accuracy was 100% if you write that, the histology of the tru-cut biopsy was glandular atypia, and the histology from the definitive surgery was carcinoma. This is just the scientific and methodological issue of how you define accuracy. The most important is the clinical 100% usefulness of the core needle biopsy because clinically the management of both glandular atypia and carcinoma would be the same.

Response:

  1. I would encourage the Author to discuss more the accuracy of the histology obtained by the tru-cut biopsy as compared with the histology obtained from definitive surgery. It is a clinically very important issue. The reported 100% accuracy is great, however, one should elaborate on this issue more. The 100% accuracy may be attributable to the meticulous preparation and double-checking of the obtained material, as presented by the Authors in the manuscript (e.g. three samples (not 1-2), the ROSE protocol). It would be reasonable to compare results with such research like Gynecol Oncol 2011;161:845-851 (that Authors already cite) or J Clin Med 2022;11:2534 or those not so recent e.g. by Zikan and Fischerova.

Response: Thanks for this comment. Discussion added

  1. Line 259: The abbreviation of the FNAC was already explained earlier in the text (line 49), so you do need to repeat it here.

Response: thanks for this comment. Deleted as suggested

  1. Line 337: “found seen cases”? It seems like a language mistake.

Response: Sorry for this mistake. Corrected

  1. Given the fact that some remarks are about English language issues, and me not being a native English speaker, I would recommend an editing of the text by a native English speaker.

Response: The text has been reviewed by a native English speaker, but he is not a scientist.

  1. In conclusion, I consider the manuscript very interesting, scientifically well conducted, and written, with additional high practical and educational value. I recommend acceptance of the manuscript after responding to the above-listed remarks. 

Response: Thanks for this comment

Round 2

Reviewer 2 Report

As a result of the changes made, the article is significantly improved.

The study presents educational value, clinical utility and is scientifically well conducted.  

The limitations associated with the small number of patients remain, as the absence of a control group to support the proposed procedures. 

The use of ROSE gives it a novelty and makes this experience worthy of publication.